# Adsorption of Non-Steroidal Anti-Inflammatory Drugs on Alginate-Carbon Composites—Equilibrium and Kinetics

**DOI:** 10.3390/ma15176049

**Published:** 2022-09-01

**Authors:** Małgorzata Wasilewska, Anna Deryło-Marczewska

**Affiliations:** Department of Physical Chemistry, Institute of Chemical Sciences, Faculty of Chemistry, Maria Curie-Sklodowska University, Maria Curie-Sklodowska Sq. 3, 20-031 Lublin, Poland

**Keywords:** alginate–carbon composites, ibuprofen and diclofenac adsorption, adsorption kinetics and equilibrium, morphology, thermal stability

## Abstract

In this work, alginate–carbon composites with different active carbon content were synthesized and studied by various techniques. The obtained materials can be used as adsorbents in the processes of removing organic pollutants from water. In this study, the effect of the immobilization of activated carbon in calcium alginate was investigated. Textural properties were determined by measuring low-temperature nitrogen adsorption/desorption isotherms. The largest specific surface area was recorded for ALG_C8 and amounted to 995 m^2^/g. The morphology of alginate materials was determined on the basis of scanning electron microscopy. The adsorption properties were estimated based on the measurements of equilibrium and adsorption kinetics. The highest sorption capacities were 0.381 and 0.873 mmol/g for ibuprofen and diclofenac, respectively. The generalized Langmuir isotherm was used to analyze the equilibrium data. A number of equations and kinetic models were used to describe the adsorption rate data, including first (FOE) and second (SOE) order kinetic equations, 1,2-mixed-order kinetic equation (MOE), fractal-like MOE equation (f-MOE), multi-exponential equation (m-exp), in addition to diffusion models: intraparticle diffusion model (IDM) and pore diffusion model (PDM). Thermal stability was determined on the basis of data from thermal analysis in an atmosphere of synthetic air.

## 1. Introduction

A large civilization leap and constantly progressing urbanization caused a growing problem related to inorganic and organic pollution of the environment. Dangerous substances undoubtedly include aromatic organic compounds [1], primarily due to their high mobility in the environment and negative consequences of long-term impact on living organisms The pharmaceutical industry is considered to be one of the most dynamically developing production sectors around the world. Recently, the presence of drugs and pharmaceuticals in natural water in addition to industrial and municipal wastewater has been observed. Existing data indicate an increase in the consumption of drugs worldwide in recent years, especially OTC (over the counter) drugs, but also drugs prescribed by a doctor [2]. The most commonly used are painkillers and anti-inflammatory drugs, cold and flu medications, heart medications, and vitamins and mineral dietary supplements. Among them, non-steroidal anti-inflammatory drugs (NSAIDs) are commonly used [2,3,4,5,6,7,8,9]. These drugs are relatively soluble in water, and as a result, they easily enter the environment from the wastewater of pharmaceutical factories, municipal wastewater from hospitals, veterinary clinics, and households. Currently, they are so widely used in human and veterinary medicine that they pose an increasing threat to the environment. Drugs from the group of NSAIDs occurring in the environment are most often detected in low concentrations, which, however, can significantly affect both aquatic and terrestrial ecosystems [10,11,12]. Moreover, these substances contain aromatic rings in their structure that are difficult to degrade, which significantly affects the half-life of such drugs in the environment. Due to the high mobility of NSAIDs in the environment and the negative influence of long-term effects on living organisms, attempts to find efficient, cheap, non-toxic, and readily available adsorbents able to permanently bind therapeutic substances are of the focus of many researchers.

One of the leading research directions in modern materials engineering is the production of composite materials of defined parameters. Such materials have been recognized and used by humans for many years. The purpose of composites is to combine two or more materials into one material with new properties. The components are often characterized by contradictory features, because of which it is possible to obtain, for example, a lightweight composite with high strength.

Recently, adsorbents of natural origin, including materials obtained on the basis of alginate—a biopolymer found in marine algae, have become very popular. Alginates exhibit mucoadhesive, antiviral, and antibacterial properties, and are also biocompatible and biodegradable [13,14], which complies with the principles of green chemistry. In combination with other materials, e.g., chitosan, hydroxyapatite or activated carbon, they create materials with various properties, used in medicine, pharmacy, and environmental protection [14]. Undoubtedly, the disadvantage of pure alginate salts is their poorly developed structure [15], which translates into a low sorption capacity in relation to organic pollutants. It should be noted that the use of activated carbon in water treatment processes has been widely spread for many years [1,14,16,17,18,19]. Concurrently, the use of this material, especially when it is in a dusty form, has some technical drawbacks, for example, difficulty in recovery. The immobilization of activated carbon in an alginate-based polymer carrier results in materials with better mechanical, textural, structural, adsorptive, and thermal properties [14].

It is notable that various polymers of natural origin are used to obtain this type of composite material. In situ gelling biopolymers undoubtedly include cellulose, chitosan, alginates, chitin, collagen, gelatin, starch, hyaluronic acid, polylactanic acid, and others. Recently, among the above mentioned, alginates and chitosan are the most widely used. Current literature data indicate numerous possible applications of chitosan-based hydrogels, primarily as drug delivery systems [20,21,22] and effective adsorbents in the processes of purifying water from contaminants, especially metal ions [23,24,25,26,27], dyes [28,29,30,31,32], and drugs [33,34,35]. It should be included, however, that chitosan is less chemically stable than alginate [14]. Alginate composites, conversely, are most often tested in environmental protection as effective sorbents for organic and inorganic pollutants. Existing data indicate that these materials are most used as adsorbents for dyes [13,14,36,37] and metal ions [14,38,39]. Concurrently, the studies on removal of drugs from aqueous solutions in systems with alginate beads are rare.

The aim of this study was to obtain and investigate the properties of composites based on alginate salts and activated carbon. A series of materials with different activated carbon content was obtained. The active carbon content in individual samples was 2 g, 4 g, and 8 g. Measurements of isotherms of low-temperature nitrogen adsorption and desorption were used to determine the textural parameters of the obtained materials. The surface topography of the samples was determined by scanning electron microscopy (SEM). The thermal stability of the composites was investigated by thermal analysis measurements. However, in order to determine the adsorption properties, measurements of equilibrium and adsorption kinetics were performed for ibuprofen and diclofenac using UV-vis spectrophotometry. The equilibrium data were analyzed using the generalized Langmuir isotherm, which corresponds to adsorption on energetically heterogeneous solids. For the analysis of the obtained kinetic data, simple equations of adsorption kinetics were used: 1st and 2nd order, multi-exponential (m-exp), mixed kinetics of 1st and 2nd order MOE, considering the effects of non-ideal f-MOE and diffusion models—intra-grain diffusion model (IDM), and pore diffusion model (PDM).

Alginate-carbon composites and pure calcium alginate were obtained. A clear influence of the presence of the modifier was observed. The obtained composite materials were characterized by a spherical grain shape, a strongly developed porous structure, high thermal stability and high adsorption capacity in relation to the tested adsorbates. Additionally, alginate-carbon composites were less polar and hygroscopic than pure calcium alginate. The best properties were demonstrated by the alginate composite with the highest active carbon content. This material, in relation to the other investigated adsorbents, demonstrated the highest specific surface area and resistance to high temperatures, in addition to the highest volume and rate of adsorption of drugs selected for research. Much higher values of the adsorption values for diclofenac were also demonstrated as a result of differences in the solubility of the tested chemical compounds. Moreover, it was demonstrated that ibuprofen adsorption was faster due to the particle size.

## 2. Materials and Methods

### 2.1. Chemicals

The following reagents were used to obtain the alginate materials. The alginate sodium salt (NaALG) was obtained from Fluka (Gillingham, UK). Calcium chloride (CaCl_2_) was purchased from Sigma-Aldrich (Tokyo, Japan). The activated carbon Norit B Supra EUR was selected from the offer of the Norit company (C, Amersfoort, The Netherlands). Before proceeding with the experiment, the carbon was washed with redistilled water to attain the supernatant conductivity below 3 µs/cm. The activated carbon used was in the form of a dusty powder and its density ranged from 370 to 400 kg/m^3^. Additionally, it had a large specific surface area (1250 m^2^/g) and high temperature resistance. Detailed information about the texture, morphology, and thermal stability of Norit B Supra EUR carbon is described in Section 3.1, Section 3.2 and Section 3.5, respectively.

As adsorbates, non-steroidal anti-inflammatory drugs were used in the research. Ibuprofen (IBP) was purchased from Fluka (Mumbai, India), Diclofenac (D) was obtained from Sigma (Beijing, China). The basic parameters characterizing the physicochemical properties of selected adsorbates are collected in Table 1.

### 2.2. Material Preparation

Spherical calcium alginate material and the alginate-carbon composites based on a calcium alginate biopolymer matrix were obtained by the gelation method. They were formed as a result of the exchange of cations of sodium alginate salt dropped into the gelling bath, a calcium chloride solution. To the reaction mixture, various amounts of the ascending modifier in the form of powdered activated carbon were added.

In the applied procedure, 150 cm^3^ of sodium alginate (NaALG; 8 g/L) were placed into four beakers, then 50 cm^3^ of redistilled water was introduced along with 2 g (ALG_C2), 4 g (ALG_C4), and 8 g (ALG_C8) of activated carbon. For one sample, no modifier was added, and pure calcium alginate (ALGCa) was obtained (reference material). The contents of the beakers were thoroughly mixed and placed into glass burettes. Concurrently, 400 cm^3^ of calcium chloride (0.075 mol/dm^3^) were measured into 8 Erlenmeyer flasks. The conical flask with the calcium chloride solution was placed on a magnetic stirrer (MR 2002, Heidolph, Schwabach, Germany). A fixed volume (75 cm^3^) of sodium alginate along with the suspended modifier was added dropwise to the CaCl_2_ solution while stirring the contents of the flask at a speed of 350 rpm. Then, the magnetic bar was removed, and the flask was protected with parafilm and left for 24 h for complete gelation. After this time, the content of the Erlenmeyer was filtered on a Büchner funnel and washed with redistilled water. The alginate composite spheres were transferred to filter paper and separated using spatulas. All steps of synthesis were performed at room temperature. The diagram of the above procedure for the preparation of alginate-based materials is presented in Figure 1.

### 2.3. Methods

#### 2.3.1. Nitrogen Adsorption/Desorption Studies

To determine the textural properties of the obtained alginate materials and the used activated carbon, low-temperature measurements of nitrogen adsorption/desorption isotherms were performed using the ASAP 2020 surface and porosity analyzer (Micromeritics, Norcross, GA, USA). Adsorption isotherms were determined at −196 °C in a wide range of relative pressures (approx. 10^−3^–approx. 0.99). The mass of the samples was 0.1 g. Before starting the analysis, the samples were degassed twice—initially for 24 h at the temperature of 60 °C in an external degassing station, and then for 12 h in the ASAP apparatus. Based on the measured nitrogen adsorption/desorption isotherms, the basic textural parameters of the adsorbents were estimated, which were: specific surface area, S_BET_, from the BET equation, external surface area, S_ext_, and mesopore volume, V_mes_, total pore volume, V_t_, from the adsorption size at relative pressure p/p_o_~0.98, micropore volume, V_mic_, t-plot method, and mean hydraulic pore size from the relationship dh = 4 V/S_BET_. The pore size distributions (PSD) were obtained by using the Barrett, Joyner, and Halenda (BJH) procedure [41].

#### 2.3.2. SEM

Scanning electron microscopy (SEM) was used to determine the morphology of the obtained materials. SEM micrographs were made using a Quanta 3D FEG high-resolution scanning electron-ion microscope (FEI, Hillsboro, OR, USA) operating at a voltage of 5 kV. Prior to measurement, the samples were placed on aluminum stub and sputtered with gold.

#### 2.3.3. Adsorption Equilibrium

Ibuprofen and diclofenac adsorption equilibrium studies were performed by using a static procedure. For this purpose, 8 series of weights (10 portions each) of the tested alginate materials were prepared, the adsorbents were transferred to Erlenmeyer flasks and contacted with aqueous solutions (pH = 7) of the adsorbates selected for testing. The systems prepared in this way were shaken for 7 days in a shaker with an incubator (New Brunswick Scientific, Edison, NJ, USA) at a temperature of 25 °C and a speed of 110 rpm. After this time, the solutions were decanted, and then absorption measurements were made using redistilled water as a reference (UV-vis Cary 4000 spectrophotometer, Varian Inc., Belrose, Australia). Spectra were recorded in the range of 200–800 nm. The maximum absorption was observed at 222 nm and 276 nm for ibuprofen and diclofenac, respectively. Relative adsorption (converted to adsorbent mass) was calculated based on material balance as:(1)$$a_{\mathrm{eq}}=\frac{\left(c_{0}-c_{\mathrm{eq}}\right)\cdot V}{m}$$
where a_eq_—relative adsorption, c_0_—initial concentration, c_eq_—equilibrium concentration, V—solution volume, and m—mass of adsorbent [1,42,43].

The obtained equilibrium data was analyzed using the generalized Langmuir isotherm (GL), which is also known as the Marczewski–Jaroniec (MJ) isotherm. This equation can describe adsorption from solutions on energetically heterogeneous solids, and has the following form:(2)$$\theta ={\left[\frac{{\left(\bar{K}\cdot c_{\mathrm{eq}}\right)}^{n}}{1+{\left(\bar{K}\cdot c_{\mathrm{eq}}\right)}^{n}}\right]}^{\frac{m}{n}}$$
where θ—relative adsorption (surface coverage), θ = a/a_m_, m, n—heterogeneity parameters (0 < m, n ≤ 1), and K—the adsorption equilibrium constant related to the characteristic energy of the energy distribution function. In special cases, this relationship may take the form of the generalized Freundlich isotherm (GF; n = 1), Langmuir–Freundlich (LF; m = n), Tóth (T; m = 1), and Langmuir (L; m = n = 1) [1,44,45].

#### 2.3.4. Adsorption Kinetics

The adsorption kinetics were measured using the UV-vis Cary 100 spectrophotometer (Varian Inc., Belrose, Australia). For this purpose, samples (0.05 g) of appropriate adsorbents were prepared and contacted with aqueous solutions of ibuprofen and diclofenac. The initial concentration of the adsorbate solutions was 0.298 mmol/L. The adsorption process was performed in a thermostatic vessel [Ecoline RE 207 thermostat (Lauda, Lauda-Königshofen, Germany)] at a temperature of 25 °C. The solution was mixed with a digitally controlled mechanical stirrer (IKA, Grójec, Poland) at a speed of 110 rpm. Changes in the concentration of the tested adsorbate solutions were monitored by recording the entire spectra in the range of 200–800 nm. The research was performed over a wide period of time, as evidenced by the number of experimental points for individual kinetic curves. A flow-through cuvette was used in the research, enabling cyclical sampling in a closed system.

Adsorption rate data were analyzed through use of many equations and models of adsorption kinetics. The first order kinetic equation (FOE)/the pseudo first order equation (PFOE) was initially used. It has a linear form, which can be represented as follows:(3)$$\ln\left(c_{\mathrm{eq}}-c\right)=\ln(c_{\mathrm{eq}}-c_{o})-k_{1}$$
or
(4)$$\ln\left(a_{\mathrm{eq}}-a\right)={\mathrm{lna}}_{\mathrm{eq}}-k_{1}t$$
where c—the temporary concentration, a—the actual adsorbed amount, the “o” and “eq” indices signify the initial and equilibrium values, and k_1_—the adsorption rate coefficient [1,46,47]. 

Another simple model is the second order equation (SOE)/the pseudo second order equation (PSOE), which can be expressed in the following form:(5)$$a=a_{\mathrm{eq}}\left[k_{2}t/\left(1+k_{2}t\right)\right]$$
or its linear forms:(6)$$t/a=\left(1/a_{\mathrm{eq}}\right)\left(1/k_{2}+t\right)$$
and
(7)$$a=a_{\mathrm{eq}}-\left(1/k_{2}\right)\left(a/t\right)$$
where k_2_ = k_2a_a_eq_ and k_2a_ are the rate coefficients for pseudo-second order kinetics [1,47,48,49]. 

The 1,2-mixed-order kinetic equation (MOE) was also applied to analyze adsorption kinetic data. This relationship is a generalization of the first and second order kinetics. It corresponds mathematically to the linear combination of the first and second order units. This dependence can be expressed as a relative adsorption progress, F, in time:(8)$$F=a/a_{\mathrm{eq}}=\frac{1-\exp\left(-k_{1}t\right)}{1-f_{2}\exp\left(-k_{1}t\right)}$$
or
(9)$$\ln\left(\frac{1-F}{1-f_{2}F}\right)=-k_{1}t$$
where f_2_ < 1—the normalized share of the second order process in the kinetics. It should be noted that under special conditions the MOE equation simplifies to the first order (f_2_ = 0) and the second order (f_2_ = 1) kinetic equations [50,51,52].

The fractal-like MOE equation (f-MOE) was also used to analyze the adsorption rate data. This dependence considers the nonideality effects. It has the following form:(10)$$F=\frac{1-\exp{\left(-k_{1}t\right)}^{p}}{1-f_{2}\exp{\left(-k_{1}t\right)}^{p}}$$
where p—the fractal coefficient. In special cases this equation is reduced to the MOE (p = 0), f-FOE (f_2_ = 0), and f-SOE (f_2_ = 1) equations [1,42,52,53].

The multi-exponential equation (m-exp) was also applied to description of kinetic data. It is commonly used for adsorption on heterogeneous solids. This dependence can bring closer a series of the first order processes or the follow up processes. It can be formulated as:(11)$$c=\left(c_{0}-c_{\mathrm{eq}}\right)\sum_{i=1}^{n}f_{i}\exp\left(-k_{i}t\right)+c_{\mathrm{eq}}$$
or
(12)$$c=c_{0}-c_{0}u_{\mathrm{eq}}\sum_{i=1}^{n}f_{i}\left[1-\exp\left(-k_{i}t\right)\right]$$
where “i”—the term of m-exp equation, k_i_—the rate coefficient, u_eq_ = 1 − c_eq_/c_0_—the relative loss of adsorbate from the solution [1,19,42,48].

In the further part, diffusion models were also used. Initially, the intraparticle diffusion model (IDM) proposed by Crank was applied. This dependence characterizes the adsorption processes on the spherical adsorbent grains. For experimental systems with a constant adsorbate concentration, it has the following form:(13)$$F=1-\frac{6}{\pi^{2}}\sum_{n=1}^{\infty }\frac{1}{n^{2}}\exp\left(\frac{-\pi^{2}\cdot n^{2}\cdot D_{a}\cdot t}{r^{2}}\right)$$
where r—the radius of adsorbent particle, D_a_—the effective diffusion coefficient: $D_{a}=\frac{D}{\tau_{p}\cdot \left(1+\rho \cdot K_{H}\cdot \varepsilon_{p}\right)}$, where: D—the molecular diffusion coefficient, τ_p_—the dimensionless pore tortuosity factor, ρ—the particle density, ε_p_—the particle porosity, K_H_—the Henry adsorption constant.

In the case when the concentration of the adsorbate is variable, then we achieve:(14)$$F=1-6\cdot \left(1-u_{\mathrm{eq}}\right)\sum_{n=1}^{\infty }\frac{\exp\left(-p_{n}^{2}\cdot \frac{D_{t}}{r^{2}}\right)}{9\cdot u_{\mathrm{eq}}+{\left(1-u_{\mathrm{eq}}\right)}^{2}\cdot p_{n}^{2}}$$
where p_n_—the non-zero roots of the equation: $\tan p_{n}=\frac{3p_{n}}{\left[3+\left(\frac{1}{u_{\mathrm{eq}}}-1\right)p_{n}^{2}\right]}$ [1,42,54].

Finally, the received kinetic data were analyzed by means of the pore diffusion model (PDM), which was proposed by McKay. It relates to the adsorption on a porous solid. This model assumes the resistance to the transport of adsorbate molecules through the surface layer, proportional penetration of the adsorbate into the adsorbent grains, and a sharp boundary between the space where the equilibrium state is determined and the space without adsorbate. It can be mathematically expressed as:(15)$$\frac{\mathrm{dF}}{{d\tau }_{s}}=\frac{3\left(1-u_{\mathrm{eq}}\cdot F\right)\cdot {\left(1-F\right)}^{\frac{1}{3}}}{1-B\cdot {\left(1-F\right)}^{\frac{1}{3}}}$$
where τ_s_—the undersized model time, u_eq_—the relative adsorbate loss, the parameter B = 1 − 1/Bi, Bi = K_f_/D_p_—the Biot number, D_p_—the pore diffusion coefficient, K_f_—the external mass transfer coefficient, and $\tau_{s}=\frac{1}{6\cdot u_{\mathrm{eq}}}\left\{\left(2B-\frac{1}{b}\right)\cdot \ln\left[\left|\frac{x^{3}+X^{3}}{1+X^{3}}\right|\right]+\frac{3}{a}\ln\left[\left|\frac{x+X}{1+X}\right|\right]\right\}+\frac{1}{X\cdot \sqrt{3u_{\mathrm{eq}}}}\cdot \left\{\mathrm{arc}\tan\left(\frac{2-X}{X\cdot \sqrt{3}}\right)-\mathrm{arc}\tan\left(\frac{2\cdot x-X}{X\cdot \sqrt{3}}\right)\right\}$, where: $x={\left(1-F\right)}^{\frac{1}{3}}$, $b={\left(\frac{1}{1-u_{\mathrm{eq}}}\right)}^{\frac{1}{3}}$ [1,42,55].

#### 2.3.5. Thermal Analysis

The thermal analysis investigation was performed by using the QMS 403D Aelos cooperating with mass spectrometer STA449F1 Jupiter (Netzsch, Selb, Germany) and TGA-IR Tensor 27 (Bruker, Billerica, MA, USA). The research materials were placed in an alumina crucible and heated at a speed 10 K/min in wide the temperature range 303–1223 K. These tests were performed under synthetic air atmosphere with a flow rate of 25 mL/min. The gaseous products emitted during decomposition for pure alginate materials and experimental systems after adsorption of ibuprofen and diclofenac on ALG_C8 were analyzed by quadrupole mass spectroscopy. The QMS data in the range from 10 to 200 amu were registered.

## 3. Results

### 3.1. Nitrogen Adsorption/Desorption Studies

Low-temperature nitrogen sorption measurements were performed for alginate-carbon composites and for comparative purposes for pure calcium alginate and the used activated carbon. Based on the conducted experiments, nitrogen adsorption/desorption isotherms were obtained, the pore volume distribution functions according to their diameter were determined by the method of Barret, Joyner, and Halenda (BJH) using the adsorption and desorption data, and the basic parameters of the structure were estimated.

Figure 2a portrays the nitrogen sorption isotherms for the tested materials. Considering the presented data, much stronger adsorption was observed for alginate-carbon composites and activated carbon in relation to calcium alginate. Among the examined composite materials, the highest adsorption was recorded for the sample with the highest active carbon content. Moreover, based on the analysis of the data portrayed here, it can be concluded that the shape of the nitrogen adsorption/desorption isotherms for samples ALG_C2, ALG_C4, ALG_C8, and C classifies them as IUPAC type I and partly IV (capillary condensation in mesopores at higher relative pressures), which is characteristic for microporous adsorbents with a share of mesoporous. The H4 hysteresis loop (according to IUPAC) is characteristic for pores with narrow gaps formed between two planes [56,57].

Figure 2b–g portrays the pore volume distribution functions according to their diameters obtained by the method of Barret, Joyner, and Halenda (BJH) using adsorption (Figure 2b–d) and desorption data (Figure 2e–g). Considering the data presented here, it can be concluded that in the case of the materials ALG_C2, ALG_C4, ALG_C8, and C, the pore size distributions are similar for the tested samples. Micropores of average size about 2 nm dominate, and small mesopores with a diameter of 3 to 4 nm are also found. In the case of ALGCa, the dominance of larger pores with a diameter of 5 nm to 30 nm was recorded.

In Table 2 the basic textural parameters of the investigated materials (ALG_C2, ALG_4, ALG_8) and for the comparative purposes of calcium alginate and activated carbon, are summarized. Based on the analysis of the data presented here, a clear differentiation of the texture of the tested solids is observed. It is obvious that the addition of the activated carbon changes systematically the texture parameters of ALG_C2, ALG_4, and ALG_8 in relation to pure ALGCa. It was observed that the specific surface area and the total pore volume of the obtained adsorbents increased significantly with increasing active carbon content. Additionally, it should be noted that micropore volume is at least 30% of the total. Moreover, it was observed that the composite materials with respect to ALGCa have about 4 times smaller mean hydraulic radius (d_h_). Concurrently, the value of d_h_ increases with the increase in the amount of activated carbon in the obtained adsorbents, however, the differences are not significant. For all composites it is about 2.5 nm. It is evident that the activated carbon forms the porous structure of the materials obtained. Depending on the need to create complex materials for various applications, strict control of the number of components of different natures is essential to design the final structure of adsorbents of desirable characteristics.

### 3.2. SEM

The effect of the addition of activated carbon on the surface topography and the grain size of alginate materials was investigated using scanning electron microscopy (SEM). Figure 3 portrays the SEM micrographs of pure ALGCa (Figure 3a–d), ALG_C8 (Figure 3e–h) (which shows the best structure parameters), and of pure C for comparative purposes (Figure 3i–l). Considering the data presented here, it can be concluded that pure calcium alginate is characterized by a spherical grain shape with a diameter of about 0.8 mm and a smooth surface. Additionally, it was observed that the alginate–carbon composite ALG_C8 also existed in the form of spherical grains. These grains were much larger and their diameters were about 2.5 mm. Moreover, the strongly developed structure of this adsorbent is clearly visible. Moreover, based on the analysis of the SEM micrographs for the activated carbon used in the synthesis, it was noted that this material was dusty, characterized by an irregular grain shape, the largest of which had a diameter of about 0.08 mm. The data obtained from scanning electron microscopy confirmed the conclusions drawn from the data of the nitrogen adsorption/desorption measurements. Based on the analysis of SEM micrographs, it was confirmed that the calcium alginate spheres were smooth, which translated into a very small specific surface area. In the case of the alginate–carbon composite, the surface irregularities and pore development of the ALG_C8 grains was clearly observed, which corresponded to large specific surface area and pore volume.

### 3.3. Adsorption Equilibrium

To estimate the adsorption effectiveness of the obtained alginate–carbon composites, the measurements of ibuprofen and diclofenac removed from aqueous solutions were performed. In Figure 4 the isotherms (Figure 4a,b), adsorption capacities (Figure 4c), scheme of adsorption mechanism (Figure 4d), and the dependence of sorption capacity on solubility (Figure 4e) for the investigated experimental systems are presented. For comparative purposes, the equilibrium adsorption data for the selected drugs on pure calcium alginate are also presented.

The received experimental data of the adsorption equilibrium were analyzed by using the generalized Langmuir equation, the parameters of which are listed in Table 3. In the case of ibuprofen adsorption on composite materials for all studied systems, the heterogeneity parameter n is equal to 1, which means that the GL isotherm is reduced to the generalized Freundlich equation. The same situation was observed for the adsorption of diclofenac on ALG_C4 and ALG_C8. For only the experimental systems IBP/ALGCa, D/ALGCa and D/ALG_C2, the full form of the generalized Langmuir isotherm equation was used to describe the adsorption equilibrium. Based on the data summarized in Table 3, the average heterogeneity effect was recorded for all studied research systems. Moreover, the corrected values of the adsorption capacities for individual cases are similar to the values estimated experimentally. The analysis of the s values of standard deviations SD (a) (from 0.034% to 0.902%) and the determination coefficients R^2^ (from 0.962 to 0.990) confirmed the very good quality of the fit.

Based on analysis of the data provided in Figure 4a–c, for tested non-steroidal anti-inflammatory drugs, a pronounced increase in adsorption value is noticed along with an increase in the content of activated carbon in the alginate material. The highest adsorptions are found for samples with ALG_C8, and the lowest for ALG_C2. This effect caused dissimilarity in the structural and surface properties of the studied alginate–carbon materials. One should remember here that an increase in the specific surface area and pore volume promotes adsorption. It should be noted that due to the increasing content of the modifier in the alginate-carbon composite, the surface characteristics of the obtained materials change, which also significantly influences the adsorption efficiency of ibuprofen and diclofenac. As the carbon content in the adsorbent increases, its hydrophobic character increases, which favors adsorption of aromatic organic compounds relatively soluble in water. The presence of activated carbon results in the occurrence of intermolecular interactions, such as π-π stacking. In the case under consideration, π electrons from the active carbon graphene layers and π electrons from the aromatic ring of drug molecule participate in the formation of these interactions [1]. Additionally, it should be noted that the adsorption studies were performed for aqueous solutions of ibuprofen and diclofenac. This means that in the tested systems, one should expect the presence of competitive adsorption between the adsorbate molecules and the solvent molecules. Therefore, in the analyzed adsorption systems, the surface groups of alginate–carbon composites may form hydrogen bonds with both functional groups of drugs and water molecules. Moreover, it should be added that under given experimental conditions (pH = 7), ibuprofen and diclofenac molecules were dissociated and electrostatic attraction forces may be expected. In Figure 4d a scheme of the adsorption mechanism of selected drugs on alginate–carbon materials is presented. Based on the analysis of data presented in Figure 4a–c, higher amount of adsorption was observed for diclofenac with respect to ibuprofen. The recorded maximum adsorption values were within the ranges of 0.257–0.381 and 0.521–0.873 mmol/g, respectively, for IBP and D (Figure 4c). This dependence can be correlated based on the differences in the water solubility of these substances [1,19,43,58]. In the case of the tested non-steroidal anti-inflammatory drugs, ibuprofen is more soluble in water than diclofenac. This indicates that among the aromatic organic compounds chosen for studies, D is more hydrophobic, which in turn points to its intensified affinity for the hydrophobic surface of alginate–carbon composites, which favors greater adsorption. In Figure 4e the dependencies of adsorption capacity as a function of the solubility in water of the adsorbates are presented. Based on the analysis of the data presented here, it is clearly visible that there is a strong, almost linear dependence between the adsorption capacity and the solubility value. The tendency is similar for the tested alginate–carbon composites. Parallel relationships were practically observed.

### 3.4. Adsorption Kinetics

In addition to the adsorption equilibrium studies, in order to complete the information on the effectiveness of the obtained carbon–alginate composites, adsorption kinetics were also investigated. For characterization of adsorbent effectiveness, the sorption capacity is very important, but also the rate at which the desired substances absorb is of great importance.

In Figure 5 the kinetic profiles for all composites are presented. Considering the data presented in Figure 5a,c, it is visible that the loss of drug concentration from the solution increases with increasing content of activated carbon in the adsorbent, the highest being recorded for systems with ALG_C8, and the lowest for ALG_C2. Furthermore, by analyzing the data contained in Figure 5b,d, the largest adsorption for ALG_C8 is also provided. Based on the analysis of Figure 5e–j, it is demonstrated that for all investigated systems, greater loss of adsorbate concentration from the solution was observed for ibuprofen. Simultaneously, grater relative adsorption was recorded for the systems with diclofenac. As already mentioned, diclofenac is less soluble in water and has an intensified affinity for the hydrophobic surface of alginate–carbon composites, thus higher adsorption values are found for this compound. However, ibuprofen particles are much smaller than diclofenac, which means that they diffuse faster to the adsorbent surface and have easier access to the porous structures of the tested composites.

A detailed analysis of the kinetics of ibuprofen and diclofenac adsorption on alginate-carbon materials was started with the compilation of experimental data in linear Bangham coordinates (Figure 6). The above procedure enables the preliminary identification of the sorption mechanism [59]. Based on the analysis of the presented data, it is observed that the plots are almost linear (slope~1) and their initial slopes, in a time interval from 10 to 1000 min, are 0.76–0.94 (IBP) and 0.79–0.96 (D), which indicates that the pure IDM mechanism (slope~0.5) cannot be used [22]. Therefore, equations considering the linear dependence of the initial rate over time should be preferred. In the next stage of interpretation, the received experimental data of the adsorption rate were analyzed by applying many equations and models of adsorption kinetics, the relative standard deviations of which are summarized in Table 4. The analysis of the data collected in Table 4 clearly indicated that the use of the multi-exponential equation made it possible to obtain the best quality of fit for the kinetic data. The parameters of the m-exp equation are listed in Table 5.

Based on the analysis of the data in Table 5, it can be observed that the adsorption of chosen non-steroidal anti-inflammatory drugs on the studied alginate–carbon composites is a complicated process, the rate of which can be practically described by two terms of the multi-exponential equation. Simultaneously, only in the case of the IBP/ALG_C2 sample, three terms of the m-exp equation were applied to describe the rate of this process, but it should be noted that this third term is small; f_3_ = 0.052. The determined values of the logarithms of the rate constantly confirm the faster adsorption of IBP in relation to D. Moreover, the analysis of this parameter for individual terms of the m-exp equation for all tested experimental systems indicates that the initial adsorption stage is slower than the later one. In the initial phase of the adsorption process, the adsorbate molecules diffuse to the adsorbent surface, and then the drugs penetrate its structure, which, among other things, results in swelling of the composite grains. This, in turn, facilitates the penetration of ibuprofen and diclofenac into the porous structures of the alginate–carbon materials tested. The analysis of the value of the relative parameter of the loss of the adsorbate from the solution u_eq_, only in the case of ibuprofen adsorption confirms that the largest amount of adsorption was observed on the ALG_C8 material, and the smallest on ALG_C2. It should be noted here that in the measurements of adsorption kinetics, the concentrations of the adsorbates were relatively low. Therefore, for the tested experimental systems with diclofenac, the parameter u_eq_ = 1, which indicated a practically complete adsorption of D from the aqueous solutions. Moreover, the adsorption half-time t_1/2_ was provided for all studied experimental systems. It is determined as the time necessary to obtain half of the concentration change. The estimated values of the half-times are 527.3–920.1 and 846.1–1259.5 min for the investigated composites with IBP and D, respectively. The above t_1/2_ values indicate higher adsorption rate for ibuprofen. It was also recorded that the values of the half-time t_1/2_ parameter decreased with the increasing content of activated carbon in composite materials, which certifies the greatest adsorption kinetics on ALG_C8 adsorbent. Additionally, in Figure 7, the correlation between the adsorption half-time and the specific surface area (Figure 7a), in addition to the half-time and the pore volume (Figure 7b), are presented. The analysis of the data presented here indicates a practically linear relationship, which confirms the significant influence of the adsorbent structure on adsorption. The quality of fit for the m-exp equation is very good, which is attested by the low values of the indetermination coefficients 1-R^2^ (in the range from 2.4 × 10^−4^ to 1.8 × 10^−2^) and low values of the relative standard deviations SD(c)/c_o_ (in the range from 0.282% to 0.976%).

### 3.5. Thermal Analysis

As part of determining the thermal properties of the obtained alginate-carbon composites, thermal analysis measurements were performed. For comparative purposes, such tests were also performed for pure calcium alginate and activated carbon. Additionally, the thermal data for ALG_C8 before and after adsorption of ibuprofen and diclofenac were analyzed. Figure 8 portrays a comparison of TG, DTG, and DSC curves for individual alginate–carbon composites, calcium alginate, and activated carbon. Additionally, Figure 9 presents a comparison of TG, DTG, and DSC curves for the obtained composite materials. Additionally, in Figure 10 the TG, DTG, and DSC curves for ALG_C8 before and after ibuprofen and diclofenac adsorption are compared. Moreover, Figure 11 presents the MS profiles of the primary gaseous degradation products of the tested samples. Conversely, in Table 6, the data estimated from the analysis of the TG, DTG, and DSC curves are summarized.

Based on the analysis of the data presented here, clear differences in thermal properties of the tested samples are observed. A multi-stage thermal degradation was recorded for all alginate materials. During the course of this process, four stages have been distinguished.

Initially, we will analyze the data for pure calcium alginate (Figure 8 and Figure 11a,b, Table 6). In the studied temperature range, four main stages of thermal degradation of ALGCa were observed, during which the total weight loss was 70.54%. The first, in the range of 30–190 °C, with a weight loss of 9.77%, is endothermic and corresponds to the removal of physically adsorbed water [60,61,62,63,64,65]. It should be added that the primary dehydration process was at 99.1 °C. Moreover, in the analyzed temperature range, the signals were recorded on the MS profiles for *m*/*z* = 17 and 18, which confirmed the water removal process. The second stage is the most intense and takes place in the temperature range 190–600 °C with a weight loss of 61.1%, and corresponds to the decomposition of calcium alginate [60,61,62,63,64,65]. The process is exothermic. It should be noted that peak maxima at 210.1, 289.3, 428.3, and 545.3 °C were recorded. Considering that ALGCa is a polymer, its degradation occurs in stages. Initially, the polymer chains are broken. Subsequently, the fragments may undergo dehydration, decarboxylation, cleavage, and condensation reactions, which may result in the formation of lower molecular weight products. Finally, the carbonaceus products are oxidized. The complexity of this stage is confirmed by the data presented on the MS profiles. In the analyzed temperature range, the *m*/*z* = 44 signal was observed, which is the most intense at 545.3 °C, which proves that the primary oxidation process of calcium alginate occurs at higher temperatures. Conversely, the signals *m*/*z* = 17 and 18 were more intense at lower temperatures of about 230 °C, which confirms the removal of water and hydroxyl groups of the sample. The third stage of thermal degradation of calcium alginate occurs in the temperature range 600–800 °C, with a weight loss of 8.61%; it is endothermic and corresponds to the formation of CaCO_3_ and partly with its subsequent decomposition to calcium oxide and carbon dioxide [60,61,62,63,64,65]. This is confirmed by the presence, in the analyzed temperature range, of a signal on the MS profile, *m*/*z* = 44 at a temperature of about 727 °C. The last stage of decomposition of calcium alginate falls in the temperature range of 800–930 °C, with a weight loss of 1.06% which corresponds to the formation of residual CaCO_3_ [60,61,62,63,64,65].

For comparison, the data of the thermal degradation of active carbon used for the synthesis of the tested composites were presented in Figure 8, Figure 9 and Figure 11c,d, and Table 6. Based on the data analysis, it was observed that the C decomposition process takes place in two stages with a total weight loss of 91.09%. The first stage was in the temperature range of 30–480 °C and corresponds to the desorption of physically bound water and the cleavage of OH groups from the surface of the carbon material [1]. The second stage was recorded in the temperature range of 480–930 °C, where an exothermic process of oxidation of the sample at the temperature of 640.7 °C was observed [1]. In the analyzed temperature range, clear signals *m*/*z* = 12 and 44 were noted, which confirm that the primary decomposition process of activated carbon occurred at the given temperature. Signals *m*/*z* = 17 and 18 were also observed on the MS profile, which likely came from surface groups of carbon material.

In the case of alginate–carbon composites, as for pure calcium alginate, the decomposition process of these materials occurs in four stages with a total loss of masses 72.63, 70.67, and 85.34% for ALG_C2, ALG_C4, and ALG_C8, respectively (Figure 8 and Figure 11e–j, Table 6). The first one is endothermic; it occurs in the temperature range 30–200 °C, with a peak maximum at 79.7, 74.9, and 90.2 °C, and a weight loss of 9.02, 6.69, and 3.74% for ALG_C2, ALG_C4, and ALG_C8, respectively. It can be correlated with the removal of physically bounded water [1,60,61,62,63,64,65] and confirmed based on the presence of *m*/*z* = 17 and 18 signals on the MS profile. The second stage, recorded in the temperature range of 200–600 °C, was exothermic and occurred with a weight loss of 58.8, 61.32, and 79.65% for ALG_C2, ALG_C4, and ALG_C8, respectively. Moreover, two main steps were observed here with peak maximums at 263.4–282.4 °C and 418.5–440.6 °C. In this temperature range, the composite decomposes, where the alginate polymer structure initially breaks down and partially degrades, followed by the primary oxidation process of the carbon material from the alginate and the activated carbon of the composite [1,60,61,62,63,64,65]. Concurrently, in the analyzed temperature range, removal of activated carbon surface groups and the water formed as a result of the decomposition of the polymer structure occurred. This was confirmed by recording on the MS profiles clear signals *m*/*z* = 17, 18, 12, and 44 recorded at 200 to 600 °C. The third stage of the decomposition of alginate-carbon composites was endothermic, in the temperature range 600–800 °C, and occurred with a weight loss of 3.06, 1.74, and 1.01%, and a peak maximum at 714.6, 659.7, and 630.1 °C for ALG_C2, ALG_C4, and ALG_C8, respectively. This could be correlated with the formation of calcium carbonate and its partial decomposition to CaO and CO_2_ [60,61,62,63,64,65]. This was confirmed by the presence of the *m*/*z* = 44 signal on the MS profile in the analyzed temperature range. The last stage of thermal degradation of alginate–carbon composites occurred in the temperature range from 800 to 930 °C and occurred with a weight loss of 1.55, 0.93, and 0.90%. At this stage, the formation of the residual calcium carbonate occurred [60,61,62,63,64,65].

In summary, the alginate–carbon composites, in relation to pure calcium alginate, were characterized by higher resistance to high temperatures. Based on the presented data, in the tested temperature range, it was observed that the total weight loss for the tested samples in the case of ALG_C2 and ALG_C4 was less than for ALGCa. In the case of ALG_C8, the weight loss was greater than for ALGCa, which results from the higher carbon content in the composite. Additionally, it was recorded that drastic weight loss for all alginate–carbon composites occurred from about 430 °C, while for pure ALGCa, a rapid weight loss occurred from about 230 °C. This means that the addition of activated carbon improved the thermal stability of the alginate material. Moreover, it should be noted that the first step of thermal degradation, which corresponded to the desorption of the physically bound water, runs with weight losses of 9.77, 9.02, 6.69, and 3.74% for ALGCa, ALG_C2, ALG_C4, and ALG_C8, respectively. It could be observed that the moisture content of the pure alginate material was higher than that of the alginate–carbon composites. This meant that the composite materials absorbed less water. Therefore, it could be concluded that the addition of activated carbon reduced the polarity and hygroscopicity of the alginate materials.

Comparing the thermal properties of all obtained alginate-carbon composites (Figure 9, Table 6), it was noted that ALG_C8 had the highest resistance to high temperatures. It should be noted that the lowest total weight loss from the sample was recorded for ALG_C4, which was practically the same as for the value recorded for ALG_C2, and the highest for ALG_C8. Concurrently, it was observed that for the composite with the highest active carbon content, the peak maximum for the main material degradation process was shifted towards higher temperatures and occurred at 440.6 °C. For ALG_C2 and ALG_C4, it was registered in 426.4 and 418.5 °C, respectively. Additionally, among the tested composites, ALG_C8 was characterized by the lowest polarity and hygroscopicity, as evidenced by the lowest weight loss in the first stage of thermal degradation of the sample.

Considering that among the obtained alginate–carbon composites the best structural, adsorptive, and thermal properties were exhibited by the ALG_C8 material, thermal analysis measurements were also made for this sample after adsorption of ibuprofen and diclofenac. The decomposition of samples with embedded drugs, like pure ALG_C8, could be broadly divided into four steps.

In the case of ALG_C8 material, after diclofenac adsorption (Figure 10 and Figure 11k,l, Table 6), the total weight loss in the tested temperature range was 92.71%. The first decomposition step monitored in the temperature range 30–200 °C was endothermic, with the peak at 92.71 °C, and corresponded to the desorption of physically adsorbed water and the initial degradation of the adsorbate [1,60,61,62,63,64,65]. In the next stage, at a temperature range of 200–600 °C, with weight loss 88.14%, the exothermic processes of further decomposition of diclofenac, dehydration, degradation of the polymer part of the composite, and subsequent oxidation of carbon products from the decomposition of alginate and activated carbon occurred [60,61,62,63,64,65]. In this temperature range, the peak maximum was observed at the temperatures of 256.1, 296.6, and 542.5 °C, of which the most intense was recorded at the highest temperature given. The above conclusions were confirmed on the MS profiles by the presence and intensity of the *m*/*z* = 12, 17, 18, 44 signals and the *m*/*z* = 35 signal, which is assigned to the chlorine present in the diclofenac molecule. The last two stages, as in the case of ALG_C8, occurred in the temperature ranges 600–800 °C and 800–930 °C, with weight losses of 0.67 and 1.26%. They could also be correlated with the formation of calcium carbonate [60,61,62,63,64,65].

In the case of the ALG_C8 sample after ibuprofen adsorption (Figure 10 and Figure 11m,n, Table 6), the total weight loss in the investigated temperature range was 91.70%. The first stage of IBP/ALG_C8 decomposition was endothermic, occurred in the temperature range 30–200 °C with peak maximum at 61.3 °C, with a weight loss of 2.27%, and corresponded to the desorption of physically bound water and partial adsorbate deposition [1,60,61,62,63,64,65]. Further on, as in the case of D/ALG_C8, in the temperature range of 200–600 °C, with mass loss 87.67%, exothermic processes of ibuprofen decomposition, dehydration and degradation of the alginate composite were recorded [60,61,62,63,64,65]. In the analyzed temperature range, the peak maximum was monitored at 282.3, 401.3, 436.9, and 493.3 °C, of which the most intense occurred above 400 °C. The last two stages, as in the case of D/ALG_C8, occurred in the temperature ranges of 600–800 °C and 800–930 °C, with weight losses of 1.75 and 0.01%. They corresponded to the formation of CaCO_3_ [60,61,62,63,64,65]. The above conclusions were maintained in accordance with the data indicated on MS profiles by the presence and intensity of the *m*/*z* = 12, 17, 18, 44 signals.

In conclusion, for the samples after drug adsorption, a greater total weight loss was observed during their heating, compared to pure ALG_C8, which was caused by the presence of adsorbed drug molecules. Moreover, a slightly greater total weight loss was recorded for the D/ALG_C8 sample, which confirmed the greater amount of diclofenac adsorption than ibuprofen on ALG_C8. Moreover, it was observed that intensive decomposition of IBP/ALG_C8 material already took place at about 400 °C, while for ALG_C8 it occurred at about 430 °C. Thus, this composite, after adsorption of ibuprofen, demonstrated less resistance to high temperatures than its pure form. Additionally, the rapid decomposition of the D/ALG_C8 sample was recorded at a temperature of about 500 °C, indicating that the material after adsorption of diclofenac had better thermal properties than ALG_C8 alone.

## 4. Discussion

In the presented work, alginate-carbon composites with different carbon content were obtained. It was demonstrated that the ALG_C8 material had the best structural, adsorption, and thermal properties. This composite was characterized by a large specific surface area of 995 m^2^/g, high resistance to high temperatures (the greatest weight loss above 440 °C), low polarity, and low hygroscopicity. Additionally, it proved to be an effective adsorbent for removing commonly used NSAIDs. The maximal sorption capacities were 0.381 and 0.873 mmol/g (otherwise 78.6 and 258.5 mg/g), respectively, for ibuprofen and diclofenac. In the literature on the subject, there are works on the preparation and characterization of alginate–carbon composites. At the same time, these materials often had much worse structural parameters and thermal properties. Additionally, such materials were tested primarily for the removal of heavy metal ions and dyes. Reports on the removal of drugs from aqueous solutions are in minority. Aziz et al. [38] prepared an alginate–carbon composite with a specific surface area of 389 m^2^/g. This material was tested by adsorption of cadmium (II) and a sorption capacity of 137 mg/g was obtained. Alsohaimi et al. [66] obtained composite beads consisting of alginate, b-cyclodextrin, gelatin, and activated carbon. This material had a S_BET_ surface area of 615 m^2^/g, and its adsorption capacity was 39.36 mg/g for the adsorption of 2,4-dichlorophenol. Moreover, the composite demonstrated poor thermal stability, as large weight losses were already observed above 100 °C. Benhouria and co-workers [13] obtained activated carbon–bentonite–alginate beads with a specific surface area of 185 m^2^/g and a sorption capacity against methylene blue of 756.97 mg/g. Garg et al. [67] obtained an alginate–carbon composite wherein the activated carbon was prepared from waste peanut shells. The specific surface area of this material was 142 m^2^/g. In the study, the adsorption of Direct Blue-86 was investigated and the sorption capacity was estimated at the level of 21.6 mg/g. Hassan and co-workers [68] obtained alginate-carbon composites, for the production of which they used activated carbon obtained from apricot stones. The S_BET_ was found to be 734 m^2^/g and the adsorption maximum for arsenic (V) to be 66.7 mg/g. Iqbal et al. [39] synthesized composite beads containing hydroxyapatite, activated carbon-functionalized alginate and nano-valent copper. The specific surface area of this material was 45 m^2^/g and its sorption capacity in relation to As (III) reached 39.56 mg/g. Jung et al. [69] obtained an alginate–carbon composite. Coffee grounds were the precursor to produce activated carbon. S_BET_ was 704 m^2^/g and the maximum adsorptions were 665.9 mg/g and 986.8 mg/g for Acid Orange 7 and Methylene Blue, respectively. Shim and colleagues [70] synthesized composite beads containing alginate, activated carbon and manganese (IV) oxide. The specific surface area of such material was 32 m^2^/g, and the sorption capacity was 129.24 mg/g and 76.14 mg/g for p-cresol and tylosin, respectively. Shamsudin and co-workers [71] received alginate–carbon films. S_BET_ was 35 m^2^/g and the adsorption maximum for diclofenac was 29.9 mg/g. Marrakchi et al. [72] acquired cross-linked FeCl_3_-activated seaweed carbon/MCM-41/alginate hydrogel composite. The specific surface area was 222 m^2^/g, the maximum adsorption values were 222.32 and 190.11 mg/g, respectively, for bisphenol A plasticizer and basic blue dye. Ragab et al. [73] synthesized an alginate–carbon composite with a surface area of 27.85 m^2^/g. The obtained data indicated that the maximum adsorption capacities of amoxicillin and diclofenac were 99.6% and 98.4%, respectively. Fei and co-workers [74] obtained composites made of alginate and graphene oxide. The S_BET_ of this material was 92 m^2^/g. The sorption capacity for ciprofoxacin was 86.12 mg/g. The research indicated that the primary process of thermal degradation of this composite occurred as early as 200 °C. Ionita et al. [75] obtained alginate/graphene oxide films for which the primary degradation process occurred at 200 °C. Zhao et al. [61] synthesized calcium alginate/reduced graphene oxide composites, for which the primary process of thermal degradation was also recorded at 200 °C. In the future, attention should be paid to the effectiveness of removing from water solutions various types of drugs, such as other non-steroidal anti-inflammatory drugs, antibiotics, cardiac drugs, and others.

## 5. Conclusions

Alginate materials were synthesized; these were alginate–carbon composites of different active carbon contents (2, 4, and 8 g, respectively, for ALG_C2, ALG_C4, and ALG_C8) and pure calcium alginate (ALGCa, reference sample). The specific surface area was estimated to be 451, 839, and 995 m^2^/g and the total pore volume was 0.28, 0.52, and 0.63 cm^3^/g for ALG_C2, ALG_C4, and ALG_C8, respectively. The pore size distribution functions were similar for all alginate–carbon materials, and the mean hydraulic radius was approximately 2.5 nm.

Data obtained from scanning electron microscopy confirmed that the addition of activated carbon immobilized in calcium alginate results in differentiating of surface and porosity of alginate materials. The surface of the ALGCa was smooth, and for ALG_C8, was clearly developed.

The adsorption properties of the obtained composites were determined by measuring the equilibrium and adsorption kinetics of ibuprofen and diclofenac. Generally, differences in the value and rate of adsorption of selected adsorbates were observed. The highest adsorption capacity and rate were recorded for the material with the highest active carbon content. Moreover, a higher sorption capacity in the system with diclofenac was demonstrated as a result of differences in the solubility of selected adsorbates. The faster adsorption rate was noted for ibuprofen as a result of the molecular size differences of the investigated drugs.

Thermal analysis demonstrated that the obtained alginate-carbon composites were thermally stable up to about 200 °C. Moreover, as the amount of activated carbon in the adsorbent increased, the hygroscopicity and polarity decreased. Additionally, the course of thermal degradation for ALG_C8 before and after the adsorption of IBP and D confirmed the greater efficiency of removing diclofenac from aqueous solutions.

The synthesized composite materials indicated good adsorption properties in relation to selected non-steroidal anti-inflammatory drugs. They seem to be promising adsorbents in the processes of removing organic pollutants. Furthermore, the prepared alginate–carbon composites are biodegradable and eco-friendly materials.

## Figures and Tables

**Figure 1 materials-15-06049-f001:**
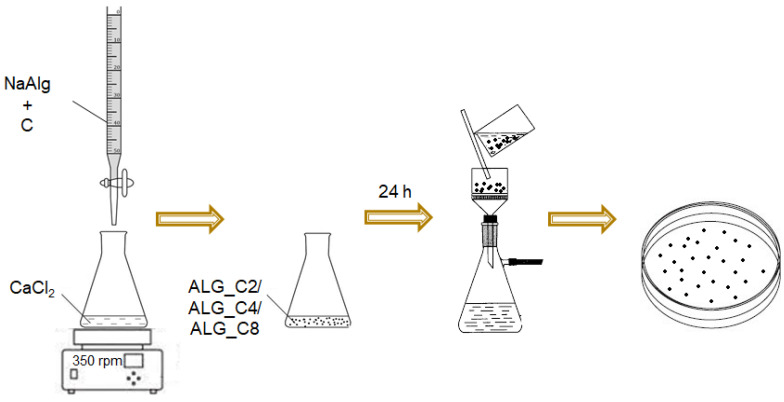
Scheme of obtaining alginate-carbon composites.

**Figure 2 materials-15-06049-f002:**
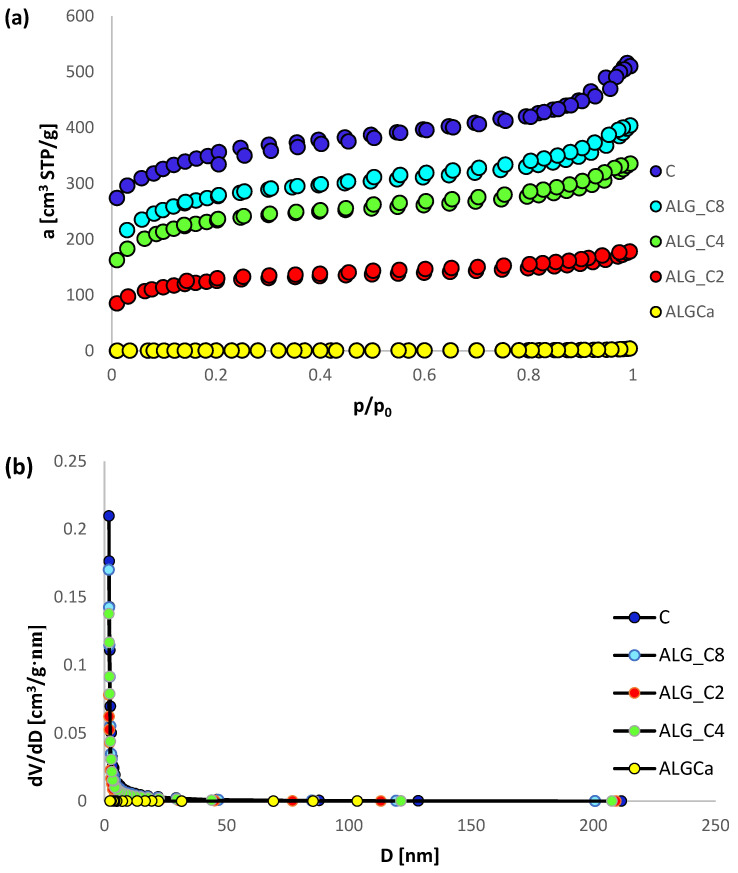
Nitrogen adsorption/desorption isotherms (**a**) pore size distributions calculated by using Barret, Joyner and Halenda (BJH) method using the adsorption (**b**–**d**) and desorption (**e**–**g**) data.

**Figure 3 materials-15-06049-f003:**
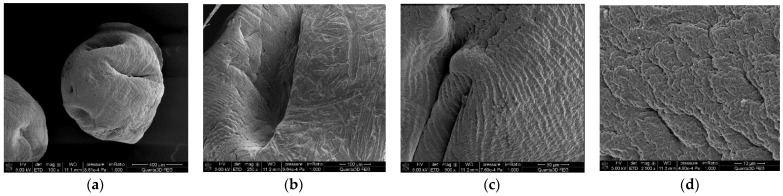
SEM micrographs ALGCa [magnifications: 100—(**a**), 250—(**b**), 500—(**c**), 2500—(**d**)], ALG_C8 [magnifications: 50—(**e**), 500—(**f**), 1000—(**g**), 5000—(**h**) and C (magnifications: 100—(**i**), 500—(**j**), 1000—(**k**), 5000—(**l**)].

**Figure 4 materials-15-06049-f004:**
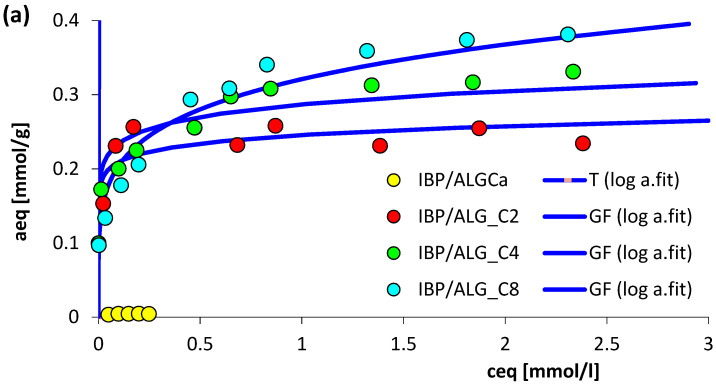
Comparison of the adsorption isotherms (**a**,**b**) (lines represent the fitted GL equation), the adsorption capacities (**c**) and proposed adsorption mechanism (**d**) for ibuprofen and diclofenac on ALGCa, ALG_C2, ALG_C4, and ALG_C8. The dependence of adsorption capacity on solubility (**e**) of ibuprofen and diclofenac on alginate–carbon materials.

**Figure 5 materials-15-06049-f005:**
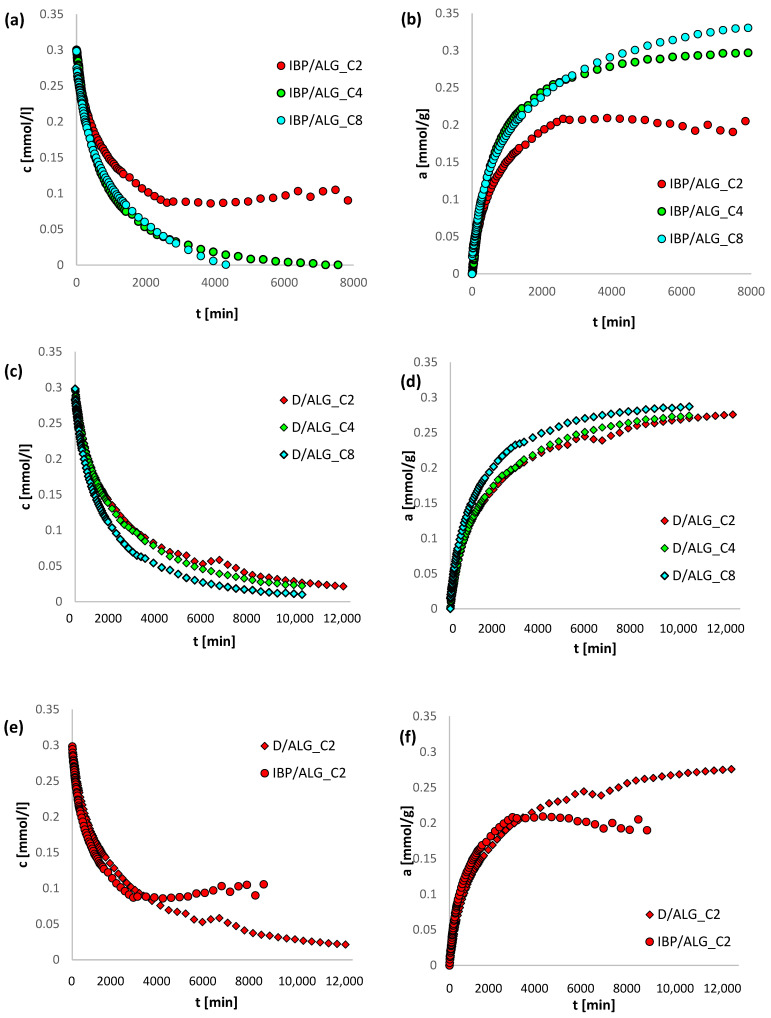
Comparison of adsorption kinetic for IBP and D on ALG_C2, ALG_C4 and ALG_C8 presented as changes in concentration (**a**,**c**,**e**,**g**,**i**) and adsorption (**b**,**d**,**f**,**h**,**j**) over time.

**Figure 6 materials-15-06049-f006:**
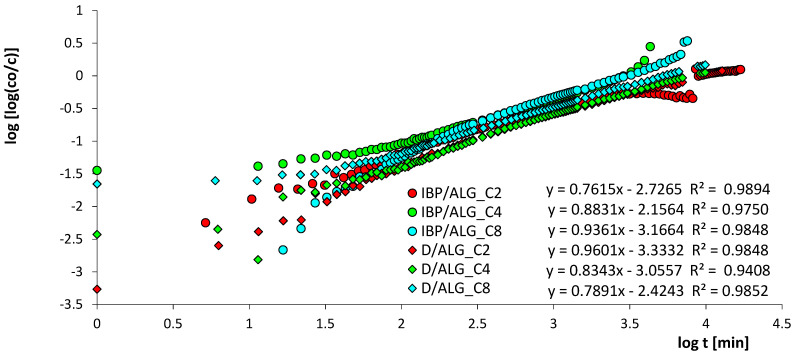
The Bangham plots for ibuprofen and diclofenac adsorption on ALG_C2, ALG_C4, and ALG_C8.

**Figure 7 materials-15-06049-f007:**
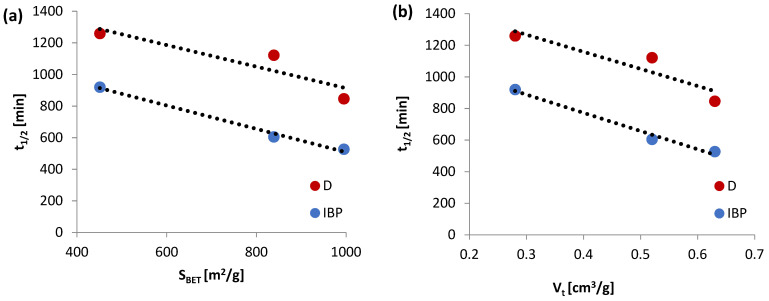
The dependence of half-times on S_BET_ (**a**) and V_t_ (**b**) for ibuprofen and diclofenac adsorbed on ALG_C2, ALG_C4, and ALG_C8 materials.

**Figure 8 materials-15-06049-f008:**
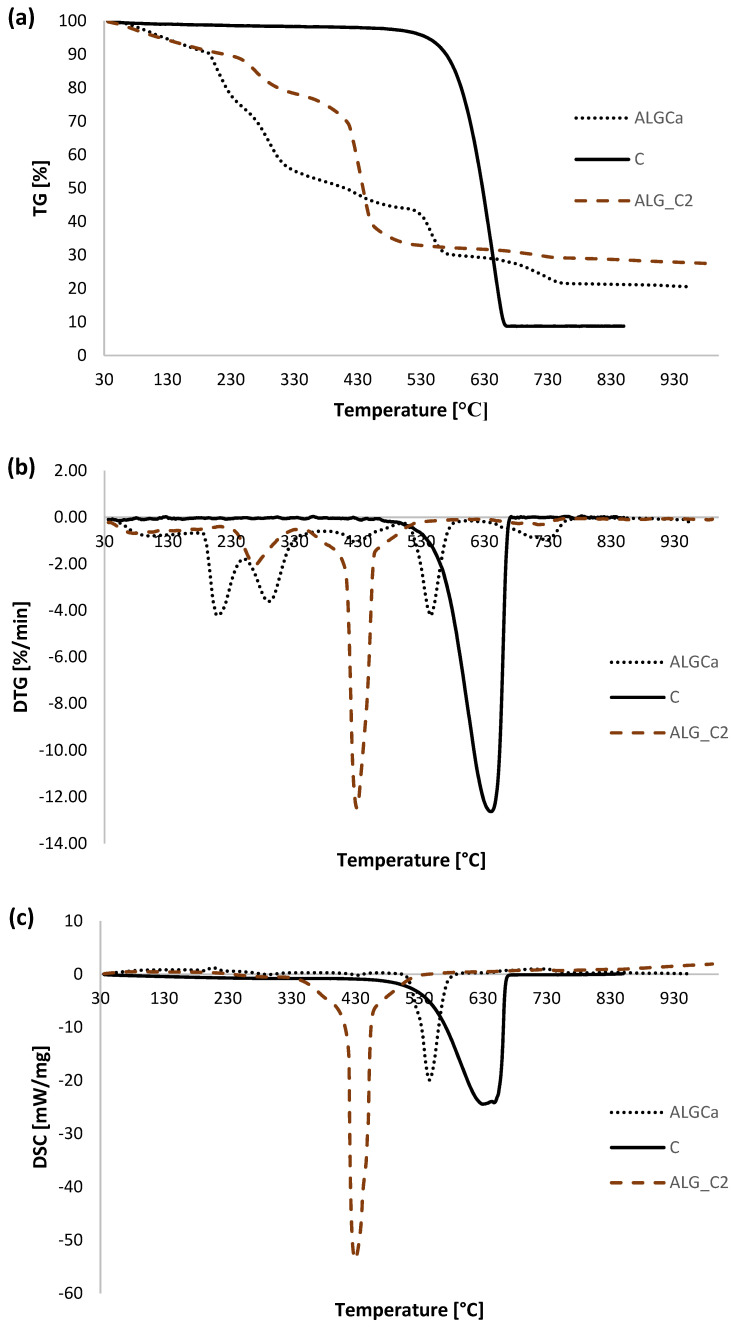
Comparison of TG (**a**,**d**,**g**), DTG (**b**,**e**,**h**) and DSC (**c**,**f**,**i**) curves for C, ALG_Ca, ALG_C2, ALG_C4, and ALG_C8 materials.

**Figure 9 materials-15-06049-f009:**
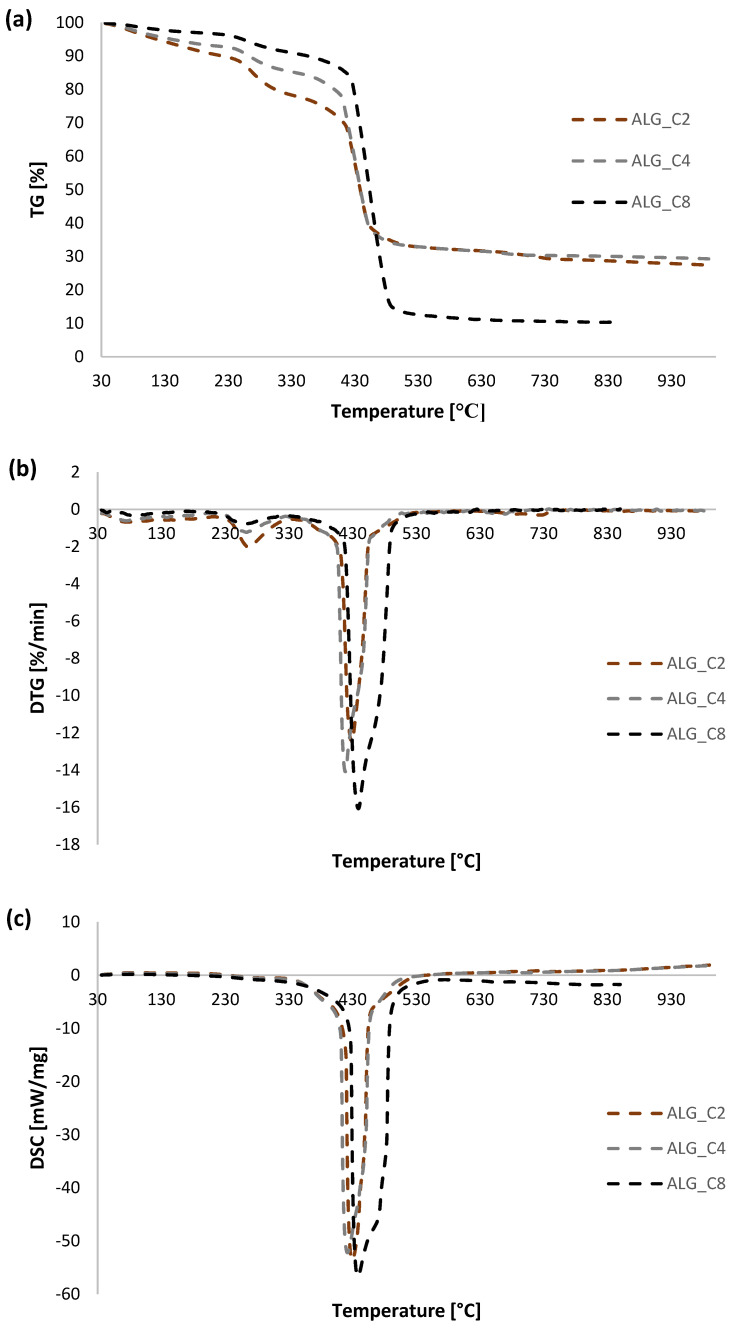
Comparison of TG (**a**), DTG (**b**) and DSC (**c**) curves for ALG_C2, ALG_C4, and ALG_C8 materials.

**Figure 10 materials-15-06049-f010:**
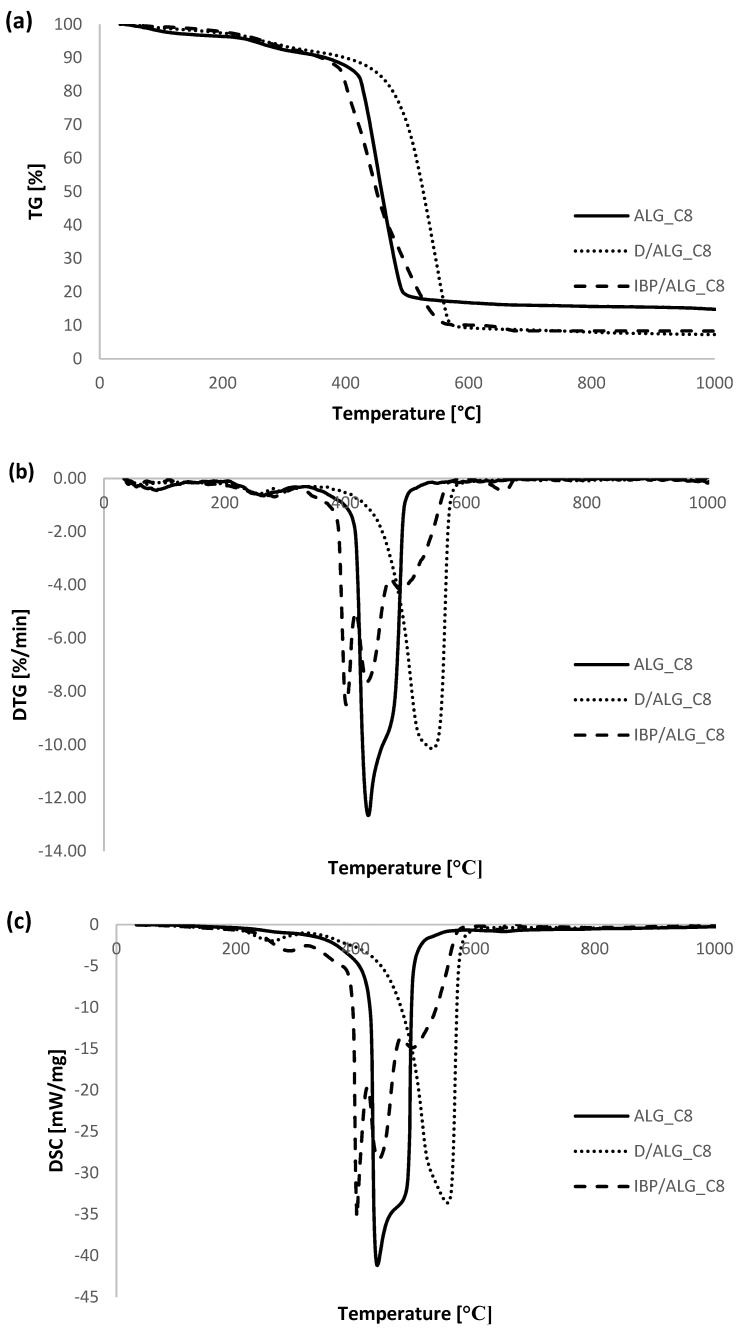
Comparison of TG (**a**), DTG (**b**), and DSC (**c**) curves ALG_C8 before and after diclofenac and ibuprofen adsorption.

**Figure 11 materials-15-06049-f011:**
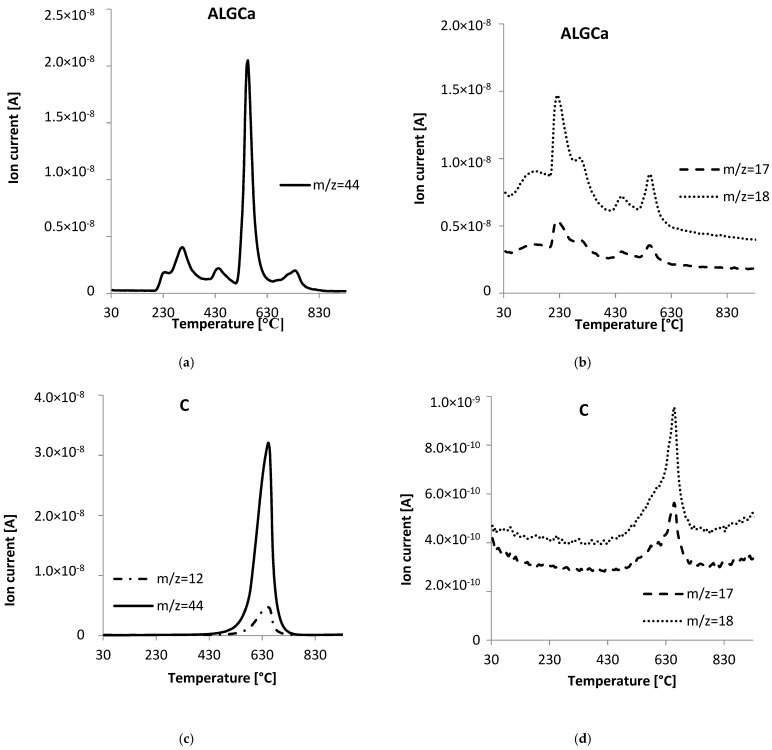
MS profiles of the primary gaseous ALG_Ca (**a**,**b**), C (**c**,**d**), ALG_C2 (**e**,**f**), ALG_C4 (**g**,**h**), and ALG_C8 (**i**,**j**) materials, in addition to ALG_C8 after diclofenac (**k**,**l**) and ibuprofen (**m**,**n**) adsorption, degradation products: OH (*m*/*z* = 17), H_2_O (*m*/*z* = 18), CO_2_ (*m*/*z* = 44), C (*m*/*z* = 12) and Cl (*m*/*z* = 35).

**Table 1 materials-15-06049-t001:** The physicochemical properties of the studied adsorbates [40].

Adsorbate	Chemical Formula	Molecular Weight [g/mol]	Water Solubility [mg/L at 25 °C]	Ionization Constant pKa	Melting Point [°C]	Boiling Point [°C]	Chemical Safety
Ibuprofen	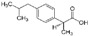	206.28	21	5.3	76	157	IrritantHealth Hazard
Diclofenac	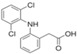	296.1	2.37	4.2	284	-	Acute ToxicIrritant

**Table 2 materials-15-06049-t002:** Textural parameters of the investigated calcium alginate, alginate-carbon composites, and active carbon.

Adsorbent	Surface Area [m^2^/g]		Pore Volume [cm^3^/g]			Pore Size [nm]	
	S_BET_ ^a^	S_ext_ ^b^	V_t_ ^c^	V_mic_ ^d^	d_h_ ^e^	BJH_ADS_ ^f^	BJH_DES_ ^g^
ALG_Ca	0.15	0.15	0.0011	-	29.33	19.70	17.16
ALG_C2	451	214	0.28	0.10	2.44	4.60	4.50
ALG_C4	839	416	0.52	0.18	2.47	4.75	4.54
ALG_C8	995	509	0.63	0.20	2.51	4.78	4.64
C	1250	632	0.80	0.26	2.55	4.94	4.77

^a^ S_BET_, BET specific surface area; ^b^ S_ext_, external surface area; ^c^ V_t_, the total pore volume; ^d^ V_mic_, the micropore volume; ^e^ D_h_, the average hydraulic pore diameter; ^f^ BJH_ads_, the BJH adsorption average pore diameter; ^g^ BJH_des_, the BJH desorption average pore diameter.

**Table 3 materials-15-06049-t003:** Parameters of the generalized Langmuir equation characterizing adsorption of ibuprofen and diclofenac from dilute aqueous solutions on calcium alginate and alginate–carbon composites.

System	Isotherm Type	a_m_	m	n	logK	R^2^	SD (a)
IBP/ALGCa	GL	0.01	0.36	0.05	0.87	0.948	0.902
IBP/ALG_C2	GF	0.28	0.69	1	−8.46	0.985	0.056
IBP/ALG_C4	GF	0.35	0.96	1	−5.94	0.986	0.047
IBP/ALG_C8	GF	0.42	0.20	1	−4.92	0.974	0.601
D/ALGCa	GL	0.01	0.38	0.05	0.91	0.962	0.813
D/ALG_C2	GL	0.52	0.87	0.80	0.90	0.984	0.046
D/ALG_C4	GF	0.64	0.48	1	0.14	0.990	0.034
D/ALG_C8	GF	0.98	0.59	1	−0.10	0.960	0.086

**Table 4 materials-15-06049-t004:** Relative standard deviations SD(c)/c_o_ (%) for m-exp, FOE, SOE, MOE, f-FOE, f-SOE, f-MOE equations, McKay pore diffusion (PDM), and IDM model (Crank).

System	m-exp [%]	FOE [%]	SOE [%]	MOE [%]	f-FOE [%]	f-SOE [%]	f-MOE [%]	IDM [%]	PDM [%]
IBP/ALG_C2	0.282	7.220	5.532	5.558	5.262	5.012	4.979	8.440	1.009
IBP/ALG_C4	0.946	5.751	7.505	0.939	5.146	6.817	1.076	10.215	2.389
IBP/ALG_C8	0.739	3.008	3.134	0.941	11.041	1.935	1.073	11.113	1.189
D/ALG_C2	0.976	3.190	1.315	0.894	1.055	1.310	0.681	15.014	1.062
D/ALG_C4	0.493	2.505	1.366	0.889	0.848	1.247	0.570	9.593	0.694
D/ALG_C8	0.879	5.426	2.045	1.095	0.703	1.806	0.651	8.731	0.344

**Table 5 materials-15-06049-t005:** Optimized parameters of m-exp equation.

System	f_1_, log k_1_	f_2_, log k_2_	f_3_, log k_3_	u_eq_	t_1/2_ [min]	SD(c)/c_o_ [%]	1-R^2^
IBP/ALG_C2	0.473, −3.728	0.475, −2.700	0.052, −1.808	0.712	920.1	0.282	1.8 × 10^−2^
IBP/ALG_C4	0.790, −3.243	0.210, −2.582	-	0.890	605.1	0.946	1.7 × 10^−2^
IBP/ALG_C8	0.540, −3.120	0.460, −2.500	-	1	527.3	0.739	3.9 × 10^−4^
D/ALG_C2	0.559, −3.934	0.441, −2.760	-	1	1259.5	0.976	8.6 × 10^−4^
D/ALG_C4	0.676, −3.489	0.324, −2.677	-	1	1122.0	0.493	2.4 × 10^−4^
D/ALG_C8	0.707, −3.339	0.293, −2.540	-	1	846.1	0.879	7.2 × 10^−4^

**Table 6 materials-15-06049-t006:** The thermal decomposition data of pure ALGCa, ALG_C2, ALG_C8, and C material as well as samples after ibuprofen and diclofenac adsorption on ALG_C8.

Sample		TG		DTG	DSC
	ΔT [°C]	Mass Loss [%]	Total Mass Loss [%]	T_min_ [°C]	endo/exo
ALGCa	30–190	9.77	80.54	99.1	endo
	190–600	61.10		210.1	exo
				289.3	exo
				428.3	exo
				545.3	exo
	600–800	8.61		726.8	endo
	800–930	1.06		-	-
C	30–480	1.99	91.09	-	-
	480–930	89.10		640.7	exo
ALG_C2	30–200	9.02	72.63	79.7	endo
	200–600	58.81		267.9	exo
				426.4	exo
	600–800	3.06		714.6	endo
	800–930	1.55		-	-
ALG_C4	30–200	6.69	70.67	74.9	endo
	200–600	61.32		263.4	exo
				418.5	exo
	600–800	1.74		659.7	exo
	800–930	0.93		-	-
ALG_C8	30–200	3.74	85.34	90.2	endo
	200–600	79.65		282.4	exo
				440.6	exo
	600–800	1.01		630.1	exo
	800–930	0.90		-	-
D/ALG_C8	30–200	2.64	92.71	65.6	endo
	200–600	88.14		256.1	exo
				296.6	exo
				542.5	exo
	600–800	0.67		-	-
	800–930	1.26		-	-
IBP/ALG_C8	30–200	2.27	91.70	61.3	endo
	200–600	87.67		282.3	exo
				401.3	exo
				436.9	exo
				493.3	exo
	600–800	1.75		660.4	endo
	800–930	0.01		-	-

## Data Availability

The data and samples are available from the authors.

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
