# Peer review of "Adsorption of Non-Steroidal Anti-Inflammatory Drugs on Alginate-Carbon Composites—Equilibrium and Kinetics"

_materials, 2022, doi:10.3390/ma15176049_

Round 1

Reviewer 1 Report

This is a well done work, presenting preparation of carbon-alginate composites for use of organic pollutant removal from aqueous solutions. After some improvements, the paper can be recommended for publication in Materials.

Major items.

1.

A major part of the work describes the analysis of the sorption kinetics using automated UV-adsorption based experiments. At various moments, step-wise irregularities can be seen in the experimental kinetic curves. Since the aim of the comparative analysis of the different models is to find out, which one fits better or worse the data, it is rather important to have accurate experimental data.

Therefore, it is recommended to repeat the experiments in several replicas, fit the data separately and only after that draw conclusions.

2.

lines 324-326: it should be explained in more details, how the SEM data confirm the nitrogen adsorption results. Or, this part may be omitted.

Minor items

Data are missing for many samples in figures 4 d and g

lines 411,412, check for some missing characters.

lines 558,559, thermal conductivity or stability?

References

The spaces and punctuation marks are somewhat careless at a few places. e.g. refs 43 and 45, first authors.

ref.44. space is missing at J.Anal.

Reviewer 4 Report

In this manuscript, the author reported the fabrication and adsorption of alginate-carbon composites for the removal of ibuprofen and diclofenac, especially for the equilibrium and kinetics. The whole manuscript was in good organization and writing, the methods was described in detail, the results is sufficient and present clearly. Therefore a minor revision of this manuscript is recommended.

(1) In the absrtact, the symbol of “FOE, SOE, MOE, IDM, PDM should be defined when it appeared first time.

(2) Some of the references in the introduction part are too old (e.g., 1997), for sure you need to update references from recently published papers (last 3 or 5 years). When introduce the adsorbents of natural origin, such as alginate and chitosan, please consider this reference as well: Gels 2022, 8, 486; Polymers 2022, 14, 2917; Materials 2022, 15, 2310.

(3) The 2.1 Materials preparation should be 2.2 Materials preparation, and 2.1 Methods should be 2.3 Methods.

(4) The physical properties of AC should be provided in the section chemicals, such as surface area, density, etc.

(5) What is the logic behind AC modification of alginate? How AC modification has improved the MB removal capability of alginate beads? Please sketch a mechanism for that.

(6) What is the initial concentration of ibuprofen and diclofenac for the adsorption?

(7) Which kinetic model is more suitable to explain the adsorption process for ibuprofen and diclofenac respectively?

Round 2

Reviewer 1 Report

in the revised version and the response letter Authors answered all issues. I suggest the paper for publication, yet, a few details can be improved as listed below.

Minor corrections

line 129. misprint in Fluka

line 208. L with capital L

line 216. use of many ….

line 223. another simple model is the second order…

lines 342-346 – please move the magnifications from here to the figure caption.

line 414. decimal point instead of comma.

line 514. attested.
